 

# Genomic and machine learning approaches to predict antimicrobial resistance in *Stenotrophomonas maltophilia*

Xin Liu,[1] Shanshan Long,[1] Fangyuan Chen,[2] Chang Liu,[3] Peng Han,[2] Hua Yu,[1] Xiaobo Huang,[4] Chun Pan,[4] Ruiming Yue,[4] Wentao Feng,[4] Guanhua Rao,[2] Han Shen,[3] Lingai Pan[4]

**ABSTRACT** *Stenotrophomonas maltophilia* is a multidrug-resistant pathogen, which poses a major challenge to clinical management due to its increasing resistance to common antibiotics, such as levofloxacin (LEV) and trimethoprim-sulfamethoxazole (SXT), and poor clinical response to treatment. There is an urgent need for rapid and reliable antimicrobial susceptibility testing (AST) methods to improve treatment outcomes. This study collected 441 *S. maltophilia* strains, performed whole-genome sequencing, and used machine learning to identify key resistance determinants for LEV and SXT, constructing predictive models for resistance phenotypes. The 441 *S. maltophilia* strains we collected show significant genomic diversity and representative lineage distribution. Machine learning identified key resistance markers for LEV and SXT, improving area under the curve values to 92.80% for LEV and 95.44% for SXT. Validation accuracies reached 94.87% for LEV and 96.27% for SXT. Mutations in parC, smeT, and gyrA were strongly associated with LEV resistance. The gene presence of sul1, sul2, and CEQ03_18740, as well as gene mutations in Gsh2, prmA, and gspD, were highly correlated with SXT resistance. These findings suggest that integrating genome-based markers can enhance the prediction of antimicrobial resistance, offering a robust method for clinical application. Genotypic AST can reliably predict resistance phenotypes, providing a promising alternative to traditional AST methods for *S. maltophilia* infections.

**IMPORTANCE** *Stenotrophomonas maltophilia* is an emerging multidrug-resistant pathogen, making treatment challenging and requiring more effective diagnostic methods. This study offers a novel approach by integrating whole-genome sequencing with machine learning to identify key resistance markers for levofloxacin and trimethoprim-sulfamethoxazole. The predictive models developed can reliably forecast antimicrobial resistance phenotypes, providing a faster and more accurate alternative to traditional susceptibility testing. This approach not only enhances clinical decision-making but also aids in the timely administration of appropriate therapies. By identifying specific genomic markers associated with resistance, this study lays the foundation for future development of personalized treatment strategies, addressing the growing concern of antibiotic resistance.

**KEYWORDS** antimicrobial resistance prediction, whole-genome sequencing, machine learning, *Stenotrophomonas maltophilia*

The rise of antibiotic resistance has become a significant public health concern, particularly due to the increased prevalence of multidrug-resistant gram-negative bacteria. *Stenotrophomonas maltophilia* is a notable pathogen among these bacteria, known for causing severe infections such as bacteremia, pneumonia, and sepsis, especially in immunocompromised hosts (1). The effective management of *S. maltophilia*

**Peer Reviewers** Xiaotong Wang, Beijing Institute of Genomics Chinese Academy of Sciences, Beijing, China; Cuidan Li, Beijing Institute of Genomics, Chinese Academy of Sciences, Beijing, China

Address correspondence to Lingai Pan, panlingai2004@163.com, Han Shen, shenhan10366@sina.com, or Guanhua Rao, gh.rao@genskey.com.

Xin Liu, Shanshan Long, and Fangyuan Chen contributed equally to this article. Author order was determined in order of increasing seniority.

The authors declare no conflict of interest.

infections is challenging due to the limited number of treatment options available and its extensive resistance to multiple antibiotics, including commonly used agents like levofloxacin (LEV) and trimethoprim-sulfamethoxazole (SXT). Earlier surveillance data from the SENTRY program indicated 96.3% susceptibility among 302 isolates from 2009 to 2012 in the US and European hospitals (2), but more recent studies have reported decreased susceptibility rates—78% in Brazil, with resistant strains often showing cross-resistance to ceftazidime (100%) and levofloxacin (52%) (3). A Chinese study found that resistance to SXT increased from 29.7% to 47.1% between 2005–2009 and 2010–2014, highlighting the rapid evolution of resistance (4). The increasing resistance to first-line treatments complicates the management of *S. maltophilia* infections, underscoring the need for early diagnosis and rapid antimicrobial susceptibility testing to guide effective treatment.

Traditional antimicrobial susceptibility testing (AST) for *S. maltophilia* requires pure cultures, and the results are influenced by factors such as incubation temperature, medium, and techniques, making the process complex and time-consuming (20–24 hours) (5). Moreover, discrepancies in interpreting the inhibition zones for SXT and variability across different AST methods further complicate the determination of optimal treatment strategies. This can lead to inconsistencies and potential low-quality evidence affecting clinical decision-making (6–10).

The development of rapid and reliable alternative antimicrobial susceptibility testing technologies has shown promise in addressing these challenges. Methods leveraging whole-genome sequencing (WGS) and metagenomic sequencing combined with machine learning for phenotype prediction are emerging as viable options for clinical application (11, 12). These advanced techniques have already been applied to other pathogens such as *Klebsiella pneumoniae* (13, 14), *Acinetobacter baumannii* (15), *Escherichia coli* (16), and *Pseudomonas aeruginosa* (17), demonstrating their potential to provide quick and accurate susceptibility profiles. However, similar approaches have not yet been extensively explored for *S. maltophilia*, indicating a critical gap in the current research landscape. Investigating genome-based AST for *S. maltophilia* could significantly enhance our ability to combat this pathogen's resistance.

In this study, we aim to address this gap by collecting the genomes of 441 clinical isolates of *S. maltophilia*. We will assess the antimicrobial susceptibility profiles for LEV and SXT and employ machine learning techniques to identify key resistance determinants for these antibiotics. By constructing predictive models for resistance phenotypes based on WGS data, we hope to establish a rapid and reliable AST method for *S. maltophilia*, contributing to better management and treatment outcomes for infections caused by this challenging pathogen.

## MATERIALS AND METHODS

### Collection of *S. maltophilia* genomes

With the removal of the low-quality genomes by a customized criterion formulated by referencing the NCBI genome exclusion rules, a total of 441 whole genomes from *S. maltophilia* isolates with AST data were collected from public databases and domestic hospitals. Among these, 329 isolates were obtained from eight hospitals in five Chinese cities (Fig. 2A). The remaining 112 genomes and antibiotic susceptibility data were sourced from published literature (18–20).

### Strain identification and antimicrobial susceptibility test

*S. maltophilia* isolates were derived from various clinical specimens, which included bile, lung tissue, celiac drainage solution, ascites, fester, airway aspiration, urine, wound secretions, blood, sputum, bronchoalveolar lavage fluid, etc. The species identification of all strains was confirmed by matrix-assisted laser desorption ionization-time-of-flight mass spectrometry VITEK MS (bioMérieux, Marcy-l'Étoile, France) in the Microbiology

Laboratory of the Clinical Medical Laboratories. The antimicrobial susceptibility test results of LEV and SXT were obtained through the following systems: VITEK 2 automated microbial identification and susceptibility system of French bioMerieux, PHOENIX identification and susceptibility system of American BD Company, TDR identification and susceptibility system of Shenzhen Mindray Biomedical Electronics Co., Ltd., and DL identification and susceptibility system of Zhuhai Deere Company. LEV and SXT antibiotic susceptibility test papers were purchased from the British Oxoid company. The Kirby-Bauer disk diffusion test and other antimicrobial susceptibility reagents were certified by the National Medical Products Administration. For all isolated strains, initial detection is performed using the Kirby-Bauer disk diffusion test or the automated instrument MIC method. For strains with initial predictions that are inconsistent, the Kirby-Bauer disk diffusion test is employed for repeated antibiotic susceptibility testing to confirm the results.

## CLSI interpretive criteria for the antibacterial susceptibility test of *S. maltophilia*

The breakpoint standard is implemented in accordance with CLSI M100 S33. The details of MIC and KB samples are as follows: SXT (disk content: 1.25/23.75 µg): ≥16 mm is sensitive, 11–15 mm is intermediate, and ≤10 mm is resistant; SXT (MIC): ≤2/38 µg/mL is sensitive and ≥4/76 µg/mL is resistant; levofloxacin (disk content: 5 µg): ≥17 mm is sensitive, 14–16 mm is intermediate, and ≤13 mm is resistant; levofloxacin (MIC): ≤2 µg/mL is sensitive, 4 µg/mL is intermediate, and ≥8 µg/mL is resistant (21).

## Phylogenetic analysis

To determine the lineage branches of the *Stenotrophomonas maltophilia* complex strains, we referenced the study by Gröschel et al. (22), selecting at least two strains from each of the 23 lineages (Table S1). A phylogenetic tree was constructed using a single-copy core gene SNP matrix aligned to the representative strain 44087_C01 (GCA_900475405.1), with the Treebest maximum likelihood model (version 1.9.2; settings: nj -t mm, -b 100). Lineage branches were identified based on the tree's evolutionary relationships, and geographic origin and antibiotic susceptibility results were mapped using iTOL.

## Identification of candidate resistance genes based on genome ORFs

The candidate resistance genes or variants were identified based on genomic open reading frame (ORF) sets. The specific steps are as follows: first, obtaining a representative sequence set of non-redundant ORFs and calculating their positive predictive value (PPV) relative to resistant phenotype. Using the predicted ORF sequences from all strains' genomes generated by Prodigal (version 2.6.3, settings: -c -m -g 11 -p single), we performed sequence clustering with CD-hit (version 4.6, settings: -c 1.0 -n 5 -G 0 -AS 0 -AL 0 -aS 1.0 -aL 1.0 -d 0 -T 0) to obtain a representative sequence set of non-redundant ORFs. Concurrently, we calculated the positive consistency rate of each ORF representative sequence with respect to resistant phenotype, The formula is as follows: PPV = the number of strains where the ORF was detected and the sensitivity was resistant/total number of strains where the ORF was detected.

Second, determining the feature type of the potential resistance gene based on the homology among different ORF representative sequences and their positive predictive values. All ORF representative sequences with identity >70% and either query or subject coverage >60% were defined as belonging to the same gene. If the PPV of all representative ORFs within this gene were ≥90% by default, the gene was classified as "presence or absence" type. Conversely, if only a subset of the representative ORFs had a PPV greater than 90%, the gene was classified as a "variant" type.

Third, selecting reference template sequences for potential resistance genes. For potential resistance genes classified as "presence or absence" type, the representative ORF with the highest detection frequency within the gene is chosen as the template

sequence. For potential resistance genes classified as variant type, the representative ORF with PPV below 90% and the highest detection frequency was selected as the template sequence. Additionally, if other ORFs with PPV below 0.9 and higher detection frequency have a significant difference in similarity with the first template sequence, they can also be selected as template sequences.

Finally, the Comprehensive Antibiotic Resistance Database (CARD) database and literature-sourced genes related to LEV and SXT resistance in *S. maltophilia* were combined with the above candidate resistance genes to form a candidate ARG reference database.

## Screening resistance markers using machine learning

Blastn (version 2.9.0+, settings: -evalue 1e-5 -num_threads 6 -outfmt 0 -num_alignments 10000) was used to align genome sequences with the ARG reference database, generating m0 format files. A custom Perl program parsed the m0 files to obtain ARG alignment regions and ARG variation information (including SNPs, insertions, deletions, and frame shifts). The feature types used for model construction included two categories: the presence or absence of genes and variations within genes. Specifically, the detection of a gene or a variation in a gene is represented as 1, while its absence is represented as 0. For each feature, the positive predictive value with respect to the resistant phenotype was calculated. Features with a PPV below 0.8 or a detection frequency below 3, as well as synonymous SNP mutation features belonging to genes, were filtered out. Ultimately, a binary distribution matrix (0-1 matrix) of the detected features was generated. The number of features was determined when the CV curve of the model reached the minimum cross-validation error value. Using the LASSO regression model (R glmnet package) and 10× cross-validation, candidate LEV/SXT resistance features were ranked by importance (13). The score of the feature obtained from model training has been supplemented, as shown in Table S3. Combined with relevant literature, resistance markers were finally selected and confirmed, and the model's performance area under the curve (AUC) value was evaluated.

## Statistical analysis

Positive predictive value, negative predictive value (NPV), sensitivity, and specificity were calculated to evaluate clinical diagnostic performance. Using Wilson scoring, 95% confidence intervals (CIs) were calculated.

## RESULTS

### Strain collection and lineage diversity

A total of 441 *S. maltophilia* complex strains were collected for this study, including 112 strains from public databases and 329 strains from clinical microbiology laboratories across five different provinces in China. The metadata for these isolates, such as source, antibiotics tested, AST method used, and lineage, are presented in Table S1. Phylogenetic analysis using a maximum likelihood tree showed that the collected strains clustered similarly to the 146 strains covering 23 lineages reported by Gröschel et al. (22), encompassing 18 of those lineages, indicating high representativeness (Fig. 1). Average nucleotide identity (ANI) analysis demonstrated that strains within the same lineage often had ANI ≥ 95%, while ANI between different lineages in the *S. maltophilia* complex was generally below 95%. For example, the ancestral Sgn4 lineage showed ANI below 90% compared to others, suggesting significant genomic diversity among lineages (Fig. 2A and B; Table S2).

We further analyzed the potential resistance genes to LEV and SXT in the collected strains. The detection rates of chromosomally encoded fluoroquinolone resistance genes —*gyrA*, *gyrB*, *parC*, and *parE*—were 96.15% (424/441), 98.19% (433/441), 93.42% (412/441), and 98.19% (433/441), respectively. The detection rates of efflux pump genes and their regulatory proteins ranged from 75.2% to 99.5%, with *smeA* having the lowest

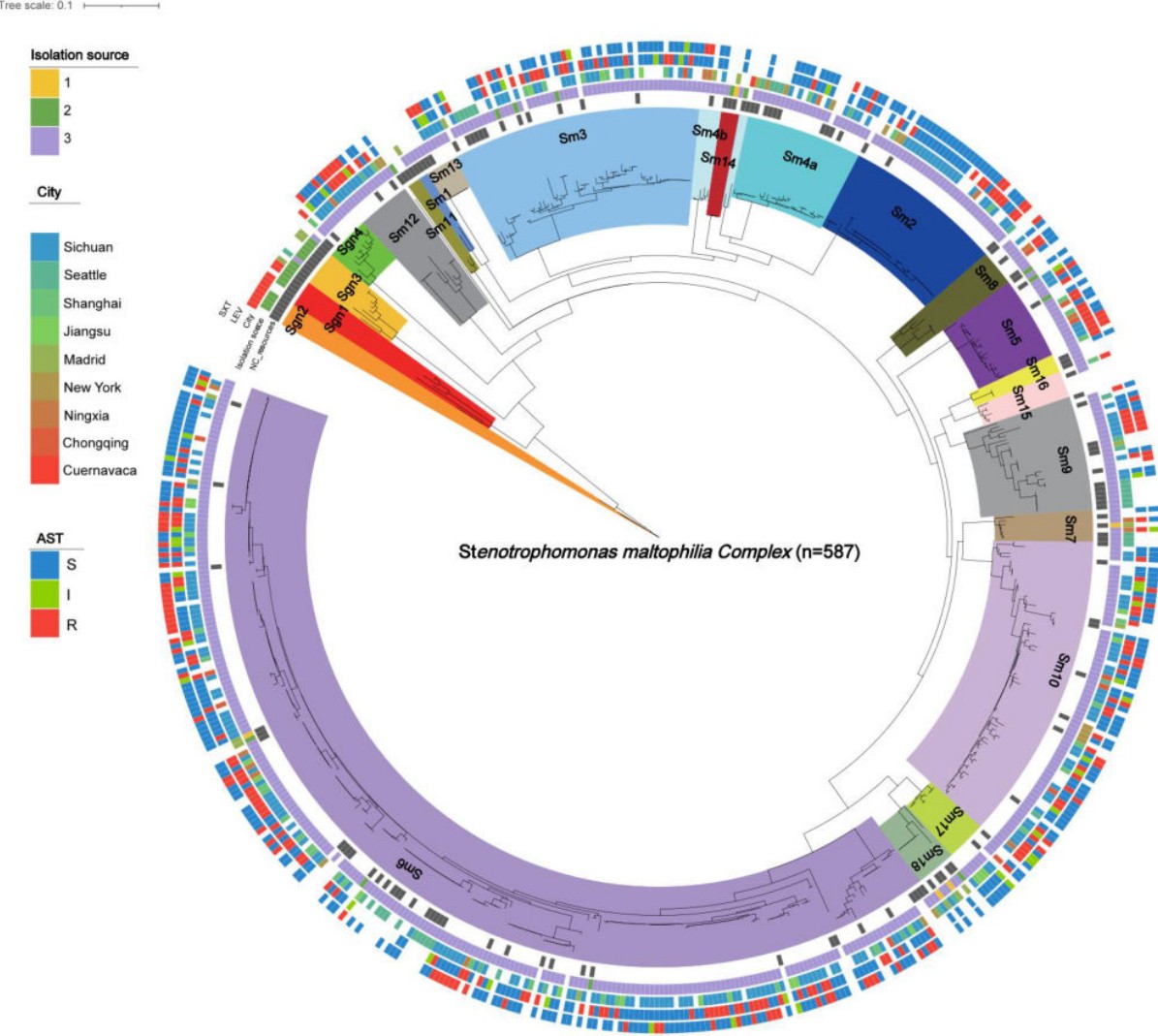

**FIG 1** Lineage distribution of the *Stenotrophomonas maltophilia* complex. Different color ranges on branches represent the 23 lineages. From inside to outside, concentric rings display (i) sources of isolates; (ii) cities of isolation; (iii) origin type (environmental, anthropogenic, or human-associated); (iv) antibiotic-resistant phenotype to LEV; and (v) antibiotic-resistant phenotype to SXT.

detection rate, particularly in lineages such as Sgn4, Sm12, Sm3, and Sm9. The Sgn4 lineage only exhibited the *smeDE* efflux pump and *smeRv* regulatory protein, differing significantly from other lineages, with an ANI below 95% (approximately 88%), indicating a distant phylogenetic relationship. The detection rates of *sul1* and *sul2* resistance genes for SXT were 7% and 2.7%, respectively, covering 33.33% (6/18) of the lineages (Fig. 2C). However, the detection of these potential resistance genes showed limited consistency with LEV and SXT resistance phenotypes, prompting us to identify more reliable genetic markers for predicting resistance phenotypes.

## Screening *S. maltophilia* resistance features using machine learning

Following our previous studies, we initially aligned the genomic sequences of the strains with a resistance feature database, correlating the detected resistance genes with antibiotic phenotypes. A Lasso regression algorithm was applied to select resistance features strongly associated with antibiotic resistance phenotypes. Using the Comprehensive Antibiotic Resistance Database (CARD), the prediction of LEV and SXT susceptibility showed high major errors and very major errors, resulting in area under the curve

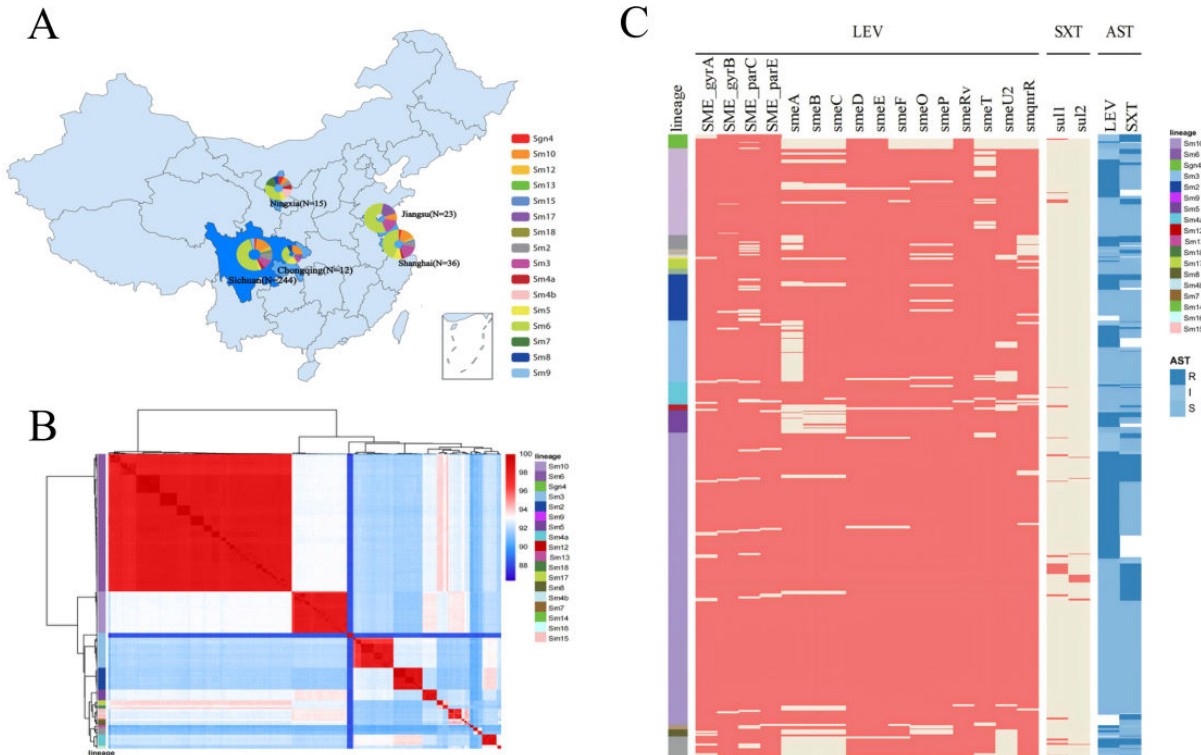

**FIG 2** Population structure of the *Stenotrophomonas maltophilia* complex. (A) Geographic origins of the 307 *S. maltophilia* isolates collected in China, with light and dark blue shades indicating isolate numbers per province. The colored pie charts show the phylogenetic lineage distribution per province. (B) Heatmap representing the clustering of pairwise average nucleotide identity among the 307 isolates, with deeper red indicating higher ANI and lighter blue indicating lower ANI. Left: distribution of phylogenetic lineages of *S. maltophilia*. (C) Phylogenetic lineage-based presence (red) or absence (yellow) of selected genes among 307 isolates, including DNA gyrase genes gyrA/gyrB, topoisomerase genes parC/parE, RND efflux pump systems smeABC, smeDEF, smeOP, efflux pump regulators smeT and smeU2, plasmid-mediated smqnrR, and dihydropteroate synthase genes sul1 and sul2. Antibiotic susceptibility results for LEV and SXT are represented in varying shades of blue.

values of only 50.0% and 62.6%, respectively (Fig. 3A and B, left panel). This indicates discrepancies between genotype and phenotype and the incomplete representation of resistance genes in CARD for *S. maltophilia*.

To address this, following the methods of Tiang et al. (16), an ORF-based candidate resistance feature screening combined with the CARD database was conducted to build a resistance feature library specific to the *S. maltophilia* complex. Using the curated *S. maltophilia* database, Lasso regression identified 23 and 17 genetic markers highly associated with LEV and SXT resistance phenotypes, respectively. In the training set, the AUC values for LEV and SXT resistance models improved to 89.34% and 86.03% (Table 1). However, the sensitivity of genotypic resistance predictions for LEV and SXT remained low (81.60% [95% CI: 74.61–87.06] and 72.32% [95% CI: 62.93–80.15], respectively). Unexpectedly, a high number of false negatives were detected for both LEV ($n = 31$) and SXT ($n = 31$). Re-testing these isolates using the Kirby-Bauer disk diffusion method showed different phenotypes from the initial AST results. Incorporating these new phenotypic results into the training set increased the AUC of genotypic resistance predictions for LEV and SXT to 92.80% and 95.44%, respectively (Table 1; Fig. 3A and B, right panel). Validation on a separate data set showed prediction accuracies of 94.87% for LEV and 96.27% for SXT. Additionally, the performance of all classifiers reaches a plateau well before utilizing the entire training set, indicating that adding more isolates for resistance classification would lead to only marginal improvements in performance (Fig. S1).

**TABLE 1** Accuracy of genotypic predictions to phenotypic AST of *Stenotrophomonas maltophilia* to LEV and SXT

| | Resistant | Intermediate[a] | Susceptible | True positive | False negative | True negative | False positive | AUC | Sensitivity (95% CI) | Specificity (95% CI) | PPV (95% CI) | NPV (95% CI) | Accuracy | Marker_num | Gene names |
|---|---|---|---|---|---|---|---|---|---|---|---|---|---|---|---|
| **LEV** (n = 430) Training set, n = 281 | 163 | 32 | 118 | 132 | 31 | 114 | 4 | 89.34% | 81.60% (74.61–87.06) | 96.61% (91.03–98.90) | 97.08% (92.23–99.06) | 79.17% (71.44–85.29) | 87.90% | 23 | parC (1), gyrA (1), smeT (2), FEO84_09045, RecT (3), LBG_14115, Fiu (4), dmlR_13 (5), K7574_18230, EmrE (6), RBl20_01405, S9_381, phnB, bdhA_1, RDM64_12445 (7), parE (1), AtoD, PaiB, prmB (8), NepI (9), NCTC10498_02179, bioF (10), RWT08_13340, |
| Re_test[b], n = 280 | 152 | 33 | 128 | 132 | 20 | 126 | 2 | 92.80% | 86.84% (80.18–91.58) | 98.44% (93.90–99.73) | 98.51% (94.17–99.74) | 86.30% (79.40–91.23) | 92.14% | | |
| Validation set, n = 117 | 15 | 0 | 102 | 10 | 5 | 101 | 1 | 82.84% | 66.67% (38.69–87.01) | 99.02% (93.88–99.95) | 90.91% (57.12–99.52) | 95.28% (88.81–98.25) | 94.87% | | |
| **SXT** (n = 405) Training set, n = 263 | 112 | 8 | 151 | 81 | 31 | 150 | 1 | 86.03% | 72.32% (62.93–80.15) | 99.33% (95.78–99.97) | 98.78% (92.45–99.94) | 82.87% (76.29–87.83) | 87.83% | 17 | sul1 (11), sul2 (11), Gsh2 (12), smeT (2), cysJ (13), 1424_2002, FEO85_09580 (14), trpS (15), BetA, NCTC10498_04164 (16), murC (17), AcrB (18), CEQ03_18740, FEO85_03140 (19), 18995_2821, prmA (20), gspD |
| Re_test[b], n = 258 | 89 | 13 | 169 | 81 | 8 | 168 | 1 | 95.44% | 91.01% (82.57–95.76) | 99.41% (96.25–99.97) | 98.78% (92.45–99.94) | 95.45% (90.93–97.87) | 96.51% | | |
| Validation set, n = 134 | 12 | 0 | 122 | 8 | 4 | 121 | 1 | 83.06% | 66.67% (35.44–88.73) | 99.18% (94.85–99.96) | 88.89% (50.67–99.42) | 96.80% (91.52–98.97) | 96.27% | | |

[a]Isolates with intermediate phenotype were excluded for model training and diagnostic performance evaluation.
[b]For strains with initial genotypic predictions that are inconsistent with phenotypic results, the Kirby-Bauer disk diffusion test is employed for repeated antibiotic susceptibility testing to confirm the results.

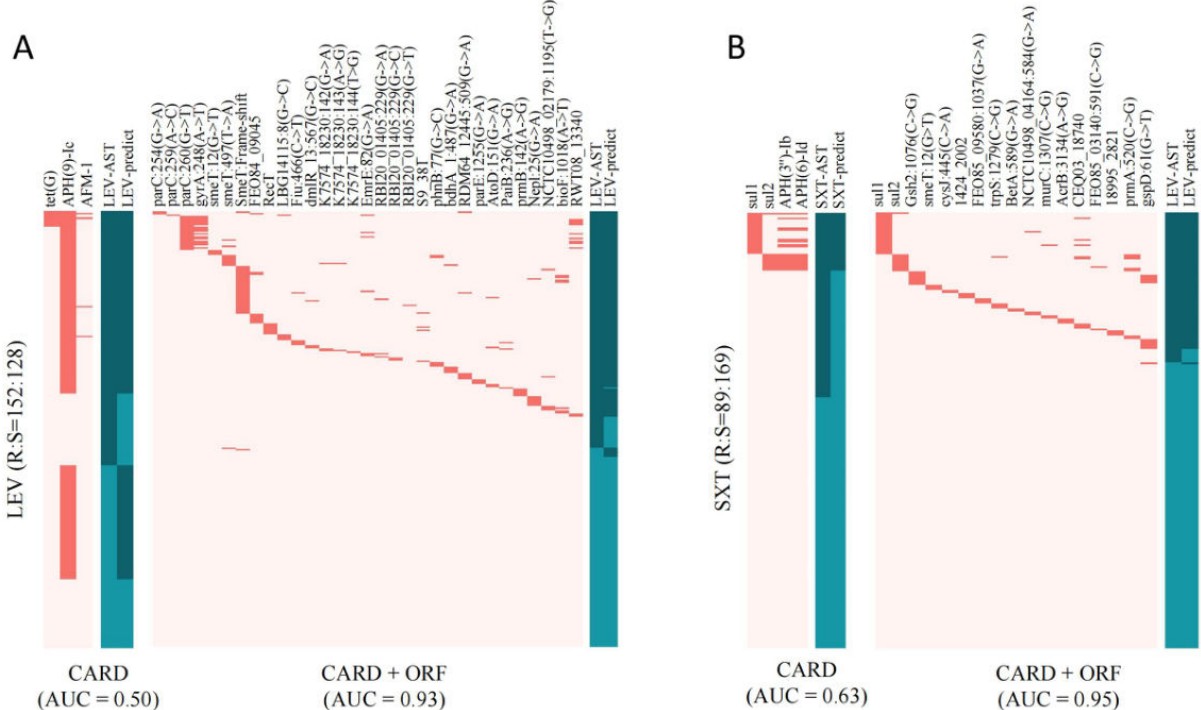

**FIG 3** Key AMR features screened for LEV and SXT resistance in *S. maltophilia* from the CARD database or the curated database (CARD plus ORF-based features). (A) LEV resistance features; (B) SXT resistance features. Left: key AMR features identified from the CARD database. Right: resistance markers identified using the ORF-based library method combined with the CARD database. Red indicates detected resistance features, and light pink indicates undetected features. Dark green represents both AST and predicted resistant phenotypes; light green represents both AST and predicted susceptible phenotypes.

## AMR determinants and resistance distribution

Gene annotation of the identified resistance sequences (Table S3) revealed that some genes previously reported to be associated with LEV resistance in Enterobacteriaceae but not in *S. maltophilia*, such as topoisomerase genes parC/E and gyrA (23), were identified in our model. Specific mutations in these genes were recognized as key resistance features linked to LEV resistance in *S. maltophilia*, such as the G-to-T mutation at position 260 in *parC*, which accounted for approximately 15% (22/152) of resistant strains. Additionally, the *smeT* gene, with mutations like G-to-T at position 12, T-to-A at position 497, and frame-shift insertions and deletions, may regulate the expression of efflux pump genes in *S. maltophilia*, affecting LEV resistance. Notably, using a single *smeT* gene template resulted in false positives for four Sgn4 lineage samples, which were corrected by adding an Sgn4-specific *smeT* gene template. For SXT resistance, apart from the *sul1* (accounting for 32.58% [29/89] of resistant strains) and *sul2* (12.36% [11/89]) genes, type II secretion system protein GspD and glutathione synthetase Gsh2 were also strongly associated with resistance. This study did not identify plasmid-mediated genes like *Smqnr*, which could potentially be related to low-level SXT resistance (24).

To explore whether the selected resistance features correlated with varying levels of LEV or SXT resistance, we plotted the distribution of these features against actual values obtained by different susceptibility testing methods (Fig. 4A through D). Some features or combinations appeared associated with high-level resistance, such as the G-to-C mutation at position 77 of the *phnB* gene in high-level LEV resistance detected by the Kirby-Bauer method (≥6 mm, Fig. 4B). Simultaneous detection of the G-to-T mutation at position 260 in *parC* and the A-to-T mutation at position 248 in *gyrA* often indicated higher resistance levels. For SXT resistance, features like *sul1* and *sul2* were only detected in high-level resistance measurements, irrespective of the Kirby-Bauer or VITEK 2 systems (Fig. 4C and D).

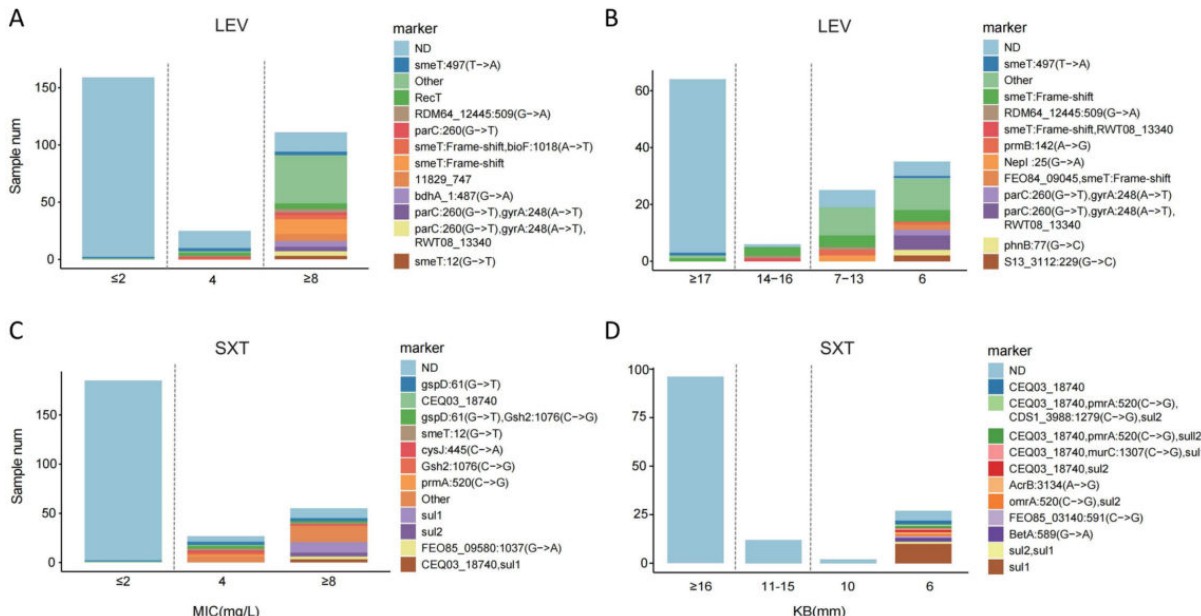

**FIG 4** Distribution of key AMR features in LEV and SXT resistance prediction models. Bar charts illustrating the distribution of LEV and SXT susceptibility testing values with the number of isolates possessing resistance features. The *y*-axis shows isolate numbers; the *x*-axis shows susceptibility values from different methods. The far-left category contains isolates with values less than or equal to the measured value, and the far-right category includes isolates with values greater than or equal to the measured value. Different colored bars represent different key resistance features. Dashed lines separate susceptible phenotypes (left), resistant phenotypes (right), and intermediate (middle). (A) LEV MIC value distribution; (B) LEV KB method distribution; (C) SXT MIC value distribution; and (D) SXT KB method distribution.

## DISCUSSION

*S. maltophilia* has been identified as the most common carbapenem-resistant gram-negative pathogen in both community and hospital-acquired bloodstream infections in the USA and ranks among the top pathogens associated with pneumonia in ICUs globally, particularly in the USA, Latin America, and Europe. Reported mortality rates for *S. maltophilia* infections vary widely, with all-cause mortality ranging from 18% to 65%, mainly derived from retrospective studies (6, 25–27). Furthermore, the rise in *S. maltophilia* infections is compounded by its complex and multifaceted resistance mechanisms, including chromosomally encoded drug efflux pumps, antibiotic-inactivating enzymes, biofilm formation, and reduced membrane permeability (1). The global spread and increasing resistance of *S. maltophilia* call for urgent improvements in diagnostic and therapeutic strategies to manage infections effectively.

This study provides critical insights into the resistance mechanisms and AST of *S. maltophilia*. We successfully identified key genetic markers associated with resistance to LEV and SXT, establishing a foundation for developing rapid, genome-based AST models that can improve patient outcomes by enabling timely and accurate antimicrobial therapy. Our study demonstrated that specific mutations in topoisomerase genes (*parC* and *gyrA*) and efflux pump regulators like smeT are closely linked to resistance phenotypes in *S. maltophilia*, suggesting that resistance mechanisms in this pathogen might differ significantly from other gram-negative bacteria (28). Notably, *S. maltophilia* is the only known bacterium where topoisomerase mutations are not traditionally linked to quinolone resistance (23). The identification of these genetic markers enhances our understanding of the complex and unique resistance pathways in *S. maltophilia* and provides a robust basis for genomic resistance prediction.

In addition to these key genetic determinants, our study employed a lineage-specific modeling approach that integrates extensive phylogenetic data, enhancing the classification and prediction of resistance phenotypes across diverse *S. maltophilia*

lineages. This approach addresses the genetic variability observed among different lineages, such as the unique efflux pump profiles of the Sgn4 lineage, which lacks many efflux pumps found in other lineages. By refining resistance predictions using Lasso regression and validating them with phenotypic AST data, we achieved significantly improved AUC values for LEV and SXT resistance prediction, reaching 92.80% and 95.44%, respectively. These performance metrics indicate the potential of our models to enhance clinical decision-making by providing more reliable AST results than traditional culture-based methods (5). Moreover, our findings underscore the potential of genomic-based predictive models as a reliable alternative to traditional AST methods, which are often subject to variability in environmental factors such as temperature and culture medium composition. The discrepancy between initial AST results and model predictions highlights the inherent challenges of current phenotypic testing, especially in the context of complex or multidrug-resistant pathogens. By integrating genomic features with machine learning, our model offers a rapid, robust, and interpretable method for predicting antibiotic resistance directly from clinical samples, significantly reducing diagnostic time. The ability to validate these predictions using supplementary phenotypic methods, such as the Kirby-Bauer disk diffusion method, further reinforces the clinical relevance of this model.

Despite these advancements, our study has limitations that warrant further exploration. The strain collection was geographically confined to six Chinese provinces, limiting the diversity and representativeness of the data. The overrepresentation of specific lineages, such as Sm6, could introduce bias in model training, necessitating broader global validation to ensure generalizability (22). Additionally, although high PPV markers were identified, these genetic features require further experimental validation to confirm their direct role in resistance. Future work should expand strain diversity and validate the findings across different clinical settings to strengthen the robustness of the predictive models. Furthermore, WGS remains a powerful tool for understanding the genetic architecture of resistance; however, its clinical implementation is still hindered by technical and cost constraints. Future work should focus on optimizing mNGS workflows to integrate resistance prediction into routine diagnostics, providing clinicians with rapid, actionable insights that can guide targeted therapy (12, 29, 30).

In summary, this study advances our understanding of *S. maltophilia* resistance mechanisms and highlights the potential of genome-based AST to transform the clinical management of infections caused by this challenging pathogen, ultimately improving treatment outcomes in vulnerable patient populations.

## ACKNOWLEDGMENTS

This work was funded by the National Key Research and Development Program of China (2023ZD0506200) and the Sichuan Provincial Science and Technology Department Key Research and Development Program (2023YFS0252).

The funder of the study had no role in the study design, data collection, data analysis, data interpretation, or writing of the article.

X.L., S.-S.L., and C.L. performed the experiments. F.-Y.C. analyzed the data and performed the data modeling. X.L., F.-Y.C., L.-A.P., G.-H.R., and W.-T.F. wrote the initial manuscript. X.L., F.-Y.C., and S.-S.L. contributed equally to this work. S.-S.L., C.L., P.H., H.Y., X.-B.H., C.P., and R.-M.Y. oversaw the acquisition of clinical and laboratory data. H.Y., H.S., X.L., and L.-A.P. provided expert advice and engaged in the data interpretation. L.-A.P., G.-H.R., H.Y., and H.S. supervised the whole project, conceptualized, and designed the study, and revised the article. All authors have reviewed the manuscript.

## AUTHOR AFFILIATIONS

[1]Department of Laboratory Medicine, Sichuan Provincial People's Hospital, University of Electronic Science and Technology of China, Chengdu, Sichuan, China
[2]Genskey Medical Technology Co., Ltd, Beijing, China

³Department of Laboratory Medicine, Nanjing Drum Tower Hospital, Nanjing, Jiangsu, China

⁴Department of Critical Care Medicine, Sichuan Provincial People's Hospital, University of Electronic Science and Technology of China, Chengdu, Sichuan, China

## AUTHOR ORCIDs

Guanhua Rao  http://orcid.org/0000-0003-4408-208X
Han Shen  http://orcid.org/0009-0002-1204-2788
Lingai Pan  http://orcid.org/0000-0002-1478-2395

## AUTHOR CONTRIBUTIONS

Xin Liu, Writing – original draft, Writing – review and editing | Shanshan Long, Writing – original draft, Writing – review and editing | Fangyuan Chen, Writing – original draft, Writing – review and editing | Chang Liu, Writing – original draft, Writing – review and editing | Peng Han, Writing – original draft, Writing – review and editing | Hua Yu, Writing – original draft, Writing – review and editing | Xiaobo Huang, Writing – original draft, Writing – review and editing | Chun Pan, Writing – original draft, Writing – review and editing | Ruiming Yue, Writing – original draft, Writing – review and editing | Wentao Feng, Writing – original draft, Writing – review and editing | Guanhua Rao, Writing – original draft, Writing – review and editing | Han Shen, Writing – original draft, Writing – review and editing | Lingai Pan, Writing – original draft, Writing – review and editing

## DATA AVAILABILITY

Whole-genome sequencing data for *S. maltophilia* are available under NCBI Bio-Project ID: PRJNA1150335. The code related to this study is referenced in our previous publication and can be accessed on GitHub (https://github.com/GoGoGao/JCM_PA_AST_model_CODE) (17).

## ETHICS APPROVAL

Ethical approvals were obtained from the Human Research Ethics Committee at the Sichuan Provincial People's Hospital (KY-N-2022-003-03).

## ADDITIONAL FILES

The following material is available online.

### Supplemental Material

**Figure S1 (Spectrum02632-24-s0001.docx).** Impact of Sample Size on LEV and SXT Resistance Model Performance.
**Supplemental material (Spectrum02632-24-s0002.pdf).** Graphical abstract.
**Supplemental tables (Spectrum02632-24-s0003.xlsx).** Tables S1 to S3.

### Open Peer Review

**PEER REVIEW HISTORY (review-history.pdf).** An accounting of the reviewer comments and feedback.

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
