## [Reviewer comments · Microbiology Spectrum]

Microbiology Spectrum

Genomic and Machine Learning Approaches to Predict Antimicrobial Resistance in *Stenotrophomonas maltophilia*

Lingai Pan, Chun Pan, Xin Liu, Shanshan Long, Fangyuan Chen, Chang Liu, Peng Han, Hua Yu, Xiaobo Huang, Ruiming Yue, Wentao Feng, Guanhua Rao, and Han Shen

Corresponding Author(s): Lingai Pan, Sichuan Academy of Medical Sciences & Sichuan Provincial People's Hospital, University of Electronic Science and Technology of China

Review Timeline:

Submission Date:	October 21, 2024
Editorial Decision:	January 24, 2025
Revision Received:	March 5, 2025
Accepted:	April 28, 2025

Editor: Fei Chen

Reviewer(s): Disclosure of reviewer identity is with reference to reviewer comments included in decision letter(s). The following individuals involved in review of your submission have agreed to reveal their identity: Xiaotong Wang (Reviewer #1); Cuidan Li (Reviewer #2)

Transaction Report:

DOI: <https://doi.org/10.1128/spectrum.02632-24>

Re: Spectrum02632-24 (**Genomic and Machine Learning Approaches to Predict Antimicrobial Resistance in *Stenotrophomonas maltophilia***)

Dear Dr. Lingai Pan:

Thank you for the privilege of reviewing your work. Below you will find my comments, instructions from the Spectrum editorial office, and the reviewer comments.

Revision Guidelines

Sincerely,
Fei Chen
Editor
Microbiology Spectrum

Reviewer #1 (Comments for the Author):

The article focuses on *S. maltophilia*, conducting comparative genomics and resistance profiling, and utilizing machine learning to develop models for identifying LEV and SXT-resistant and susceptible strains. This research aims to establish a rapid and reliable antimicrobial susceptibility testing (AST) method for *S. maltophilia*, contributing to better management and treatment outcomes for infections caused by this challenging pathogen. However, as a study centered on comparative genomics and machine learning, there are several areas for improvement:

Model Reproducibility and Generalization: With AUC values reported between 92% and 96%, the authors should upload the

model, related code, and corresponding training, testing, and validation datasets to GitHub. This would enable other researchers to replicate the results and assess the model's generalization capabilities.

Feature Selection and Importance: The authors should clarify the types and numbers of features chosen for model building, as well as provide an analysis of feature importance rankings, which are key elements in machine learning studies. Additionally, a comprehensive breakdown of selected features and their relevance to resistance prediction would be beneficial.

Phenotypic Data and Model Accuracy: In lines 207-210, the authors note that updating phenotypic test data significantly improved model accuracy. It is necessary to analyze the accuracy of various phenotypic datasets and identify the true underlying factors responsible for this increase in predictive performance.

Clarification of Novelty: Since the resistance gene screening and machine learning modeling methods based on PPV analysis has been developed in other studies, the article should clearly articulate its unique contributions.

In Methods section - PPV Analysis: While the authors mention performing PPV analysis, they do not provide details on the analysis approach. Given the importance of this method for identifying potential resistance genes, a detailed explanation is required. Additionally, the cited threshold values of 0.95 and 0.9 in lines 145-147 remain unclear in their relationship; further clarification is needed on how these thresholds were applied.

Figures: Figure 1B contains essential information but is too small and unclear. It is recommended to enlarge this figure and present it independently as a primary figure for better visibility.

The article focuses on *S. maltophilia*, conducting comparative genomics and resistance profiling, and utilizing machine learning to develop models for identifying LEV and SXT-resistant and susceptible strains. This research aims to establish a rapid and reliable antimicrobial susceptibility testing (AST) method for *S. maltophilia*, contributing to better management and treatment outcomes for infections caused by this challenging pathogen. However, as a study centered on comparative genomics and machine learning, there are several areas for improvement:

Model Reproducibility and Generalization: With AUC values reported between 92% and 96%, the authors should upload the model, related code, and corresponding training, testing, and validation datasets to GitHub. This would enable other researchers to replicate the results and assess the model's generalization capabilities.

Feature Selection and Importance: The authors should clarify the types and numbers of features chosen for model building, as well as provide an analysis of feature importance rankings, which are key elements in machine learning studies. Additionally, a comprehensive breakdown of selected features and their relevance to resistance prediction would be beneficial.

Phenotypic Data and Model Accuracy: In lines 207-210, the authors note that updating phenotypic test data significantly improved model accuracy. It is necessary to analyze the accuracy of various phenotypic datasets and identify the true underlying factors responsible for this increase in predictive performance.

Clarification of Novelty: Since the resistance gene screening and machine learning modeling methods based on PPV analysis has been developed in other studies, the article should clearly articulate its unique contributions.

In Methods section - PPV Analysis: While the authors mention performing PPV analysis, they do not provide details on the analysis approach. Given the importance of this method for identifying potential resistance genes, a detailed explanation is required. Additionally, the cited threshold values of 0.95 and 0.9 in lines 145-147 remain unclear in their relationship; further clarification is needed on how these thresholds were applied.

Figures: Figure 1B contains essential information but is too small and unclear. It is recommended to enlarge this figure and present it independently as a primary figure for better visibility.

Clarification of Novel Contributions: Since the resistance gene screening and machine learning modeling methods based on PPV analysis were developed by other studies, the article should clearly articulate its unique contributions.

Reviewer #1 (Comments for the Author):

The article focuses on *S. maltophilia*, conducting comparative genomics and resistance profiling, and utilizing machine learning to develop models for identifying LEV and SXT-resistant and susceptible strains. This research aims to establish a rapid and reliable antimicrobial susceptibility testing (AST) method for *S. maltophilia*, contributing to better management and treatment outcomes for infections caused by this challenging pathogen. However, as a study centered on comparative genomics and machine learning, there are several areas for improvement:

1. Model Reproducibility and Generalization: With AUC values reported between 92% and 96%, the authors should upload the model, related code, and corresponding training, testing, and validation datasets to GitHub. This would enable other researchers to replicate the results and assess the model's generalization capabilities.

Response: The whole genome sequencing reads data of all strains has been uploaded to NCBI database (Accession number: PRJNA1150335), and the corresponding meta-information in this study can be found in Table S1 for the training, validation, and testing sets. The code related to this study is referenced in our previous publication and can be accessed on GitHub (https://github.com/GoGoGao/JCM_PA_AST_model_CODE). We have made a statement on this in the revised manuscript.

2. Feature Selection and Importance: The authors should clarify the types and numbers of features chosen for model building, as well as provide an analysis of feature importance rankings, which are key elements in machine learning studies. Additionally, a comprehensive breakdown of selected features and their relevance to resistance prediction would be beneficial.

Response: The feature types used for model construction include the presence or absence of genes and mutations on the genes. Specifically, the detection of a gene or mutation is represented as 1, while its absence is represented as 0. The positive consistency rate (PPV) of each feature with respect to the resistance phenotype is then calculated. Features with a PPV lower than 0.8 or a detection frequency below 3, as well as synonymous SNP mutations, were filtered out. This process results in a 0-1 distribution matrix of detected features. When the CV curve of the model reached its lowest value, the number of features was determined. Correspondingly, the methods section of the article has been updated to include this description (Line 186-197). The score of features (namely feature importance ranking) obtained from model training have also been supplemented, as shown in Table S3.

3. Phenotypic Data and Model Accuracy: In lines 207-210, the authors note that updating phenotypic test data significantly improved model accuracy. It is

necessary to analyze the accuracy of various phenotypic datasets and identify the true underlying factors responsible for this increase in predictive performance.

Response: We appreciate the reviewer's comment and recognize the importance of analyzing the accuracy of various phenotypic datasets and understanding the underlying factors responsible for improvements in model performance.

As antimicrobial susceptibility testing (AST) results can be influenced by various factors, it is important to highlight that discrepancies in results may arise due to subtle variations in temperature and culture medium composition [Ref.1]. For example, in the case of *S. maltophilia*, current commercial automated testing systems demonstrate classification consistency of less than 90% across different detection systems, indicating that AST remains a challenging task. The Kirby-Bauer disk diffusion method, particularly for detecting susceptibility to Trimethoprim-sulfamethoxazole, Levofloxacin, and Minocycline, generally aligns more closely with acceptable performance standards [Ref.2]. When inconsistencies between the predictive model and AST results were observed, we opted to conduct repeat testing using the Kirby-Bauer method. For instance, the initial model predicted susceptibility for 12 levofloxacin samples that were found resistant by AST. A second validation using the Kirby-Bauer method confirmed susceptibility, thus reinforcing the model's predictive reliability.

Further, in sample 18703, a G→A mutation at position 25 of the *Nepl* gene was identified, indicating a resistance feature. While the initial levofloxacin susceptibility test showed sensitivity, the second validation confirmed resistance, increasing the positive predictive value (PPV) for this feature to 1.00 (7/7). Similarly, sample 16815 showed a G→A mutation at position 509 of the *RDM64_12445* gene, with the PPV for this feature increasing to 1.00 (6/6) after validation. A similar scenario was observed for Trimethoprim-sulfamethoxazole.

We have also added a detailed discussion of these findings in the revised manuscript (Line312-321). Thank you again for pointing out this important issue.

References:

- [1] Khan, Ayesha, et al. "Evaluation of the Vitek 2, Phoenix, and MicroScan for antimicrobial susceptibility testing of *Stenotrophomonas maltophilia*." *Journal of Clinical Microbiology* 59.9 (2021): 10-1128.
- [2] Khan, Ayesha et al. "Evaluation of the Performance of Manual Antimicrobial Susceptibility Testing Methods and Disk Breakpoints for *Stenotrophomonas maltophilia*." *Antimicrobial agents and chemotherapy* vol. 95,5 (2023): e02631-20.

Sample ID	First		Predicted		Second		
	AST_method	Result	Feature	Result	AST_method	Result	
LEV	16815	Kirby-Bauer	20(S)	R	Kirby-Bauer	16(I)	
	18703	VITEK 2	0.25(S)	R	Kirby-Bauer	11(R)	
	1529	VITEK 2	4(I)	S	Kirby-Bauer	25(S)	
	16583	VITEK 2	8(R)	S	Kirby-Bauer	15(I)	
	3126	VITEK 2	8(R)	S	Kirby-Bauer	23(S)	
	9827	VITEK 2	8(R)	S	Kirby-Bauer	19(S)	
	16949	VITEK 2	≥8(R)	S	Kirby-Bauer	26(S)	
	14557	VITEK 2	≥8(R)	S	Kirby-Bauer	25(S)	
	17708	VITEK 2	≥8(R)	S	Kirby-Bauer	28(S)	
	624	Kirby-Bauer	6(R)	S	Kirby-Bauer	19(S)	
	16137	VITEK 2	8(R)	S	Kirby-Bauer	17(S)	
	18612	Kirby-Bauer	7(R)	S	Kirby-Bauer	20(S)	
	18765	Kirby-Bauer	9(R)	S	Kirby-Bauer	22(S)	
	19005	Kirby-Bauer	6(R)	S	Kirby-Bauer	17(S)	
	20282	Kirby-Bauer	6(R)	S	Kirby-Bauer	19(S)	
	SXT	10703	Kirby-Bauer	20(S)	S	Kirby-Bauer	10(R)
		16233	VITEK 2	≥16(R)	S	Kirby-Bauer	11(I)
		71	VITEK 2	8(R)	S	Kirby-Bauer	12(I)
		13686	VITEK 2	4(R)	S	Kirby-Bauer	14(I)
17015		VITEK 2	8(R)	S	Kirby-Bauer	15(I)	
19107		VITEK 2	16(R)	S	Kirby-Bauer	15(I)	
3824		VITEK 2	4(R)	S	Kirby-Bauer	27(S)	
6479		VITEK 2	4(R)	S	Kirby-Bauer	22(S)	
10503		VITEK 2	4(R)	S	Kirby-Bauer	25(S)	
11829		VITEK 2	4(R)	S	Kirby-Bauer	17(S)	
12360		VITEK 2	4(R)	S	Kirby-Bauer	16(S)	
16718		VITEK 2	8(R)	S	Kirby-Bauer	16(S)	
405		Kirby-Bauer	6(R)	S	Kirby-Bauer	27(S)	
624		Kirby-Bauer	6(R)	S	Kirby-Bauer	28(S)	
752		Kirby-Bauer	6(R)	S	Kirby-Bauer	19(S)	
1529		VITEK 2	16(R)	S	Kirby-Bauer	28(S)	
2916		VITEK 2	8(R)	S	Kirby-Bauer	24(S)	
6063		VITEK 2	4(R)	S	Kirby-Bauer	20(S)	
11003		VITEK 2	16(R)	S	Kirby-Bauer	19(S)	
15762		VITEK 2	16(R)	S	Kirby-Bauer	22(S)	
16508		VITEK 2	4(R)	S	Kirby-Bauer	21(S)	
16583		VITEK 2	16(R)	S	Kirby-Bauer	27(S)	
PZH2		Kirby-Bauer	6(R)	S	Kirby-Bauer	22(S)	
3844		VITEK 2	16(R)	S	Kirby-Bauer	24(S)	
16493		Kirby-Bauer	6(R)	S	Kirby-Bauer	24(S)	

Result measurement unit, VITEK 2 (mg/L), Kirby-Bauer (mm).

4. Clarification of Novelty: Since the resistance gene screening and machine learning modeling methods based on PPV analysis has been developed in other studies, the article should clearly articulate its unique contributions.

Response: The methods established in our previous study have shown good performance for *Klebsiella pneumoniae* and *Escherichia coli*. However, the effectiveness for *Stenotrophomonas maltophilia* remains uncertain. This article extended the application of these methods to the *Stenotrophomonas maltophilia* species complex and had achieved the anticipated outcomes, identifying several potential new resistance genes that have not been previously reported, such as AtoD, PaiB, and BetA. More newly discovered antibiotic resistance genes can be found in Table S3.

5. In Methods section - PPV Analysis: While the authors mention performing PPV analysis, they do not provide details on the analysis approach. Given the importance of this method for identifying potential resistance genes, a detailed explanation is required. Additionally, the cited threshold values of 0.95 and 0.9 in lines 145-147 remain unclear in their relationship; further clarification is needed on how these thresholds were applied.

Response: Thanks for your valuable suggestion. We have added a more detailed description of the analysis method in the revised manuscript. Please refer to the methods section (Line 153-180): Firstly, obtaining a representative sequences set of non-redundant ORFs and calculate their positive predictive value (PPV) relative to resistant phenotype. Secondly, determining the feature type of the potential resistance gene based on the homology among different ORF representative sequences and their positive predictive

values (PPV). Finally, selecting reference template sequences for potential resistance genes. Moreover, In theory, the higher the threshold setting, the more accurate the screened features will be. However, considering the number of resistant strains and the possible inaccuracies in AST, a relatively high value of 90% was set by default. In practice, this threshold can be adjusted based on the model's performance during the construction process.

6. Figures: Figure 1B contains essential information but is too small and unclear. It is recommended to enlarge this figure and present it independently as a primary figure for better visibility.

Response: We appreciate the reviewer's suggestion to improve the visibility of Figure 1B. In response, we have enlarged the figure and presented it as a separate figure (Figure 1) to ensure better clarity and visibility. We believe this adjustment will enhance the reader's ability to interpret the data more effectively. Thank you for this valuable recommendation..

Re: Spectrum02632-24R1 (**Genomic and Machine Learning Approaches to Predict Antimicrobial Resistance in *Stenotrophomonas maltophilia***)

Dear Dr. Lingai Pan:

Your manuscript has been accepted, and I am forwarding it to the ASM production staff for publication. Your paper will first be checked to make sure all elements meet the technical requirements. ASM staff will contact you if anything needs to be revised before copyediting and production can begin. Otherwise, you will be notified when your proofs are ready to be viewed.

Sincerely,
Fei Chen
Editor
Microbiology Spectrum

Reviewer #1 (Comments for the Author):

The revised manuscript provides comprehensive responses to the reviewer's concerns, particularly regarding the screening of resistance gene-SNP variations and the selection of machine learning features. The authors have incorporated relevant textual descriptions and table data into the methods section. Additionally, they have offered a thorough discussion and explanation of specific predictive results, such as how modifying the dataset improved model accuracy. In accordance with the reviewer's suggestions, the authors have also revised the figure, enhancing the clarity of Figure 1.

Reviewer #2 (Comments for the Author):

No.

The revised manuscript provides comprehensive responses to the reviewer's concerns, particularly regarding the screening of resistance gene-SNP variations and the selection of machine learning features. The authors have incorporated relevant textual descriptions and table data into the methods section. Additionally, they have offered a thorough discussion and explanation of specific predictive results, such as how modifying the dataset improved model accuracy. In accordance with the reviewer's suggestions, the authors have also revised the figure, enhancing the clarity of Figure 1. As a whole, all the authors' conclusions supported by their data and the manuscript written in standard English and easy to comprehend.